

# Alterations to arbuscular mycorrhizal fungal community composition is driven by warming at specific elevations

Mei Yang[1,2,3], Zhaoyong Shi[1,2,3], Bede S. Mickan[4,5], Mengge Zhang[1,2,3] and Libing Cao[1]

[1] College of Agriculture, Henan University of Science and Technology, Luoyang, China
[2] Henan Engineering Research Center for Rural Human Settlement, Luoyang, China
[3] Luoyang Key Laboratory of Symbiotic Microorganism and Green Development, Luoyang, China
[4] UWA School of Agriculture and Environment, the University of Western Australia, Perth WA, Australia
[5] The UWA Institute of Agriculture, the University of Western Australia, Perth WA, Australia

## ABSTRACT

**Background:** Global warming can alter plant productivity, and community composition which has consequences for soil-plant associated microorganisms. Arbuscular mycorrhizal fungi (AMF) are distributed widely and form symbiotic relationships with more than 80% of vascular plants and play a key role in nutrient cycling processes at the ecosystem scale.

**Methods:** A simulated warming experiment at multiple elevations (3,000, 3,500, 3,800, and 4,170 m) was conducted utilizing an in-situ open-top chamber (OTC) for exploring the effect of global warming on AMF community structure in the Qinghai-Tibet Plateau (QTP). This region has been identified as one of the most sensitive areas to climatic changes. Soil DNA was extracted and sequenced using next the Mi-Seq platform for diversity profiling.

**Results:** AMF richness was higher under the simulated warming chamber, however this only occurred in the elevation of 3,500 m. Warming did not alter other AMF alpha diversity indices (e.g. Shannon, Ace, and Simpson evenness index). *Glomus* and *Acaulospora* were the dominate AMF genera as assessed through their relative abundance and occurrence in control and warming treatments at the different elevations.

**Conclusion:** Warming changed significantly AMF community. The effects of warming on AMF community structure varied depend on elevations. Moreover, the occurrences of AMF in different genera were also presented the different responses to warming in four elevations.

## INTRODUCTION

Arbuscular mycorrhizal fungi (AMF) are distributed widely and form symbiotic relationships readily with more than 80% of vascular plants (*Yang et al., 2012*; *Li et al., 2020b*; *Shi et al., 2020b*). There are many AMF benefits of the symbiosis to plant physiological, and also

Corresponding author
Zhaoyong Shi, shizy1116@126.com

broader ecological processes are influenced (*Phillips & Hayman, 1970*; *Colla et al., 2008*; *Ren et al., 2017*; *Bi, Xiao & Sun, 2019*). AMF utilize carbon in the form of photosynthate from host plants in exchange for enhanced nutrient access to the plant from the symbiosis (*Shi, Miao & Wang, 2014*). Mycorrhizal plants can also transfer more photosynthate from shoot to roots of non-mycorrhizal plants (*Marschner, Crowley & Higashi, 1997*).

AMF are important components of soil biological processes and functional links between plants and soil (*Yang et al., 2010*). Mycorrhizal fungi have a vital impact on the composition of microbial and plant communities (*van der Heijden et al., 2015*; *Genre et al., 2020*), and AMF symbiosis can also improve nutrient and water supply to host plants (*Parniske, 2008*; *Song et al., 2015*; *Wang, Pokharel & Chen, 2019*; *Sarmiento-López et al., 2020*; *Shi et al., 2020a*). Positive plant water relations by AMF have been demonstrated to improve plant drought resistance (*Chen et al., 2020*) by enhancing the uptake of N and P under drought stress (*Hashem et al., 2018*), alleviate soil water stress (*Mickan et al., 2016*), and promote plants to deplete soil moisture to alleviate plant water stress (*Hardie, 1985*). Many studies also suggested that AMF may promote plant growth through enhance tolerance to abiotic stress, such as drought and salinity (*Yang, Koide & Zhang, 2016*; *Xiang et al., 2016*; *McKibben & Henning, 2018*; *Higo et al., 2019*; *Zhang et al., 2019a*, *2019b*; *Wu et al., 2021*).

Mycorrhizas also play an important role in biodiversity of plants and ecosystem functions (*Zhao et al., 2017*; *He et al., 2010*), by influencing plant community diversity and composition (*van der Heijden et al., 1998*; *van der Heijden, 2004*; *Pagano, Cabello & Scotti, 2010*). Therefore, AMF play a fundamental role in the origin, evolution, distribution, survival, growth, and development of plants and larger ecosystem scale processes (*Liu & Wang, 2003*; *Wang & Qiu, 2006*; *McGuire et al., 2008*; *Hiiesalu et al., 2014*). AMF has an independent phylum Glomeromycota based on taxonomic status, which probably evolved from Ascomycota and Basidiomycota (*Schußler, Schwarzott & Walker, 2001*), and has an estimated 1,250 species of AMF worldwide (*Borstler et al., 2006*). *Opik et al. (2013)* analyzed the AMF community of 96 plant roots and found 59 new virtual taxa (VT), using high throughput bar coded amplicon diversity profiling. Overall, the preservation of AMF diversity is important for plant diversity, net primary productivity, and ecosystem maintenance (*Mahmoudi et al., 2019*).

With the challenges of climate change, and the influence of global warming on AMF community composition has received greater attention due to their role in larger ecosystem level processes (*Sun et al., 2013*). Under moderate temperature, there can be positive influences of AMF plant tolerance to salinity, indicating temperature is a key component to AMF related processes (*Wu & Zou, 2010*). Transferred carbon from host plants to AMF can also be temperature dependent, with reported increases below 18 °C with warming and decreases above 18 °C (*Gavito et al., 2015*). Warming directly decreased AMF colonization across plant species and across the climate gradient in prairie plants along a Mediterranean climate gradient (*Wilson et al., 2016*). Warming has also been demonstrated to reduce AMF species richness, though there were no negative effects (*Shi et al., 2017*). However, they only studied the influence of warming on AMF in single

**Table 1 The sampling sites and coordinates based on different elevations on the Qinghai-Tibet Plateau.**

| Elevation | Sample location | Longitude | Latitude |
|---|---|---|---|
| 3,000 m | Near the Redstone Observation Deck | E102°02′3.42″ | N29°50′36.49″ |
| 3,500 m | Near the Yajiageng Timber Checkpoint | E102°02′9.50″ | N29°51′42.90″ |
| 3,800 m | Near the rock | E102°01′2.30″ | N29°53′20.80″ |
| 4,170 m | Go up the Yajiageng Boundary Monument for 1 km | E102°0′42.50″ | N29°54′26.70″ |

elevation, which was probably difficult to evaluate accurately the responses of AMF to warming during climate changes.

The most sensitive region to climate changes in the world is the Qinghai-Tibet Plateau (QTP), where is a global biodiversity hotspot because it provides a natural "laboratory" for the development of natural science research with unique geographical environment (*Tian et al., 2009*; *Shi et al., 2015*). How AMF communities respond to warming at different elevations is limited on the Qinghai-Tibetan Plateau, and to this end we investigated the influence of warming on AMF community based on four elevations.

Due to the temperature-sensitive nature of the Qinghai-Tibet Plateau, we have made the following assumptions: (1) Warming significantly changes the AMF community structure. (2) Warming significantly changes AMF richness. (3) The changes of AMF community are consistent at four elevations after warming.

## MATERIALS & METHODS

### Site description

The Qinghai-Tibet Plateau is a vast plateau in Central Asia covering most of the Tibet Autonomous Region and Qinghai Province in China. Referred to as the "the roof of the world" occupying 2.5 million square kilometers, it is the highest and biggest plateau of the world. The annual average temperature is −4 °C, with annual precipitation ranges from 100 to 300 mm. In our study, the main vegetation is *Kobresia pygmea* and the type of soil is alpine meadow soil. The slope for each sampling site is less than 2°. In view of the uniqueness of climatic and topographical characteristics on QTP, this study selected samples between 29°50′36.49″–29°54′26.70″ north latitude and 102°0′42.50″–102°02′9.50″ east longitude on the eastern part of QTP (Table 1).

### Experiment design and sample collection

Quadrats of 20 m × 20 m were positioned at four elevations of 3,000, 3,500, 3,800 and 4,170 m on QTP. Each quadrat was divided into 25 of 4 m × 4 m little quadrats. We took three biological repetitions with non-adjacent randomly as control treatment (CK) and OTC warming treatment (OTC) by the way of artificial and simulated warming through open-top chamber, respectively (*Gao & Li, 2019*; *Li et al., 2020a*). Compared with other warming methods, it can ensure that the soil is basically undamaged and easily to repeated

(*Klein, Harte & Zhao, 2004*). The top and bottom are hexagonal and open with the side composed of six trapezoid-shaped plexiglass. We carried out a 1-year warming test and all samples were taken in August and September of the next year without rain or snow. Small meteorological observation stations were set up at each altitude to monitor soil temperature and moisture. Instantaneous measurement of soil temperature and soil moisture was performed by fixed-point measurement using HOBO PRO temperature and soil moisture recorder. We selected soils samples randomly with a soil corer with diameter of two cm and depth of 0–20 cm. We mixed three soil cores as a sample and repeated three times in CK and OTC, respectively. Then, separating the root system from the soil and sealing with sterile plastic valve bags, with DNA samples being stored at –20 °C. Field experiments were approved by the Key Laboratory of Mountain Surface Processes and Ecological Regulation, Chinese Academy of Sciences (20160416).

## DNA extraction and PCR amplification

Genomic DNA was extracted from soil samples, using the Fast DNA SPIN Kit for Soil (MP Biomedicals LLC, Santa Ana, CA, USA) according to manufacturer's protocols. The final DNA purification and concentration were determined by Nano Drop 2000 UV-vis spectrophotometer (Thermo Scientific, Wilmington, NC, USA), and DNA quality was checked by 1% agarose gel electrophoresis. The extracted DNA was subjected to nested PCR by thermocycler PCR system (GeneAmp 9700; ABI, Sunnyvale, CA, USA). The first PCR amplification was performed with primers AML1F (5′-ATCAACTTTCGATGGTAGGAT AGA-3′) and AML2R (5′-GAACCCAAACACTTTGGTTTCC-3′) by an ABI GeneAmp® 9700 PCR thermocycler (ABI, Sunnyvale, CA, USA) (*Lee, Lee & Young, 2008*; *Li et al., 2019*). The PCR reactions were conducted using the following program: 3 min of denaturation at 95 °C, 32 cycles of 30 s at 95 °C, 30 s for annealing at 55 °C, and 45 s for elongation at 72 °C, and a final extension at 72 °C for 10 min. PCR reactions were performed in triplicate 20 μL mixture containing 4 μL of 5 × FastPfu Buffer, 2 μL of 2.5 mM dNTPs, 0.8 μL of each primer (5 μM), 0.4 μL of FastPfu Polymerase and 10 ng of template DNA. The second PCR amplication used identical reaction conditions described above with the primers AMDGR (5'-CCCAACTATCCCTATTAATCAT-3') and AMV4-5NF (5'-AAGCTCGTA GTTGAATTTCG-3') (*Shi et al., 2019*), and the following program: 3 min of denaturation at 95 °C, 30 cycles of 30 s at 95 °C, 30 s for annealing at 55 °C, and 45 s for elongation at 72 °C, and a final extension at 72 °C for 10 min. The resulted PCR products were extracted from a 2% agarose gel and further purified using the AxyPrep DNA Gel Extraction Kit (Axygen Biosciences, Union City, CA, USA) and quantified using QuantiFluor™-ST (Promega, Madison, WI, USA) according to the manufacturer's protocol.

## Illumina MiSeq DNA sequencing

Purified barcoded amplicons were pooled in equimolar concentrations and paired-end sequenced on an Illumina MiSeq PE300 platform/NovaSeq PE250 platform (Illumina, San Diego, CA, USA) according to the standard protocols by Majorbio Bio-Pharm Technology

Co. Ltd. (Shanghai, China). The raw reads were deposited into the NCBI Sequence Read Archive (SRA) database (Accession Number: PRJNA694003).

## Processing of sequencing data

The raw sequencing reads were demultiplexed, quality-filtered by fastp version 0.20.0 (*Chen et al., 2018*) and merged by FLASH version 1.2.7(*Magoč & Salzberg, 2011*). Microbial community sequencing was conducted by Shanghai Majorbio Bio-pharm Technology using Illumina-MiSeq sequencing platform. The data were analyzed on a free online platform of Majorbio Cloud Platform (www.majorbio.com).

Operational taxonomic units (OTUs) were clustered with 97% similarity cutoff using UPARSE version 7.1, and chimeric sequences were identified and removed (*Stackebrandt & Goebel, 1994*; *Edgar, 2013*). The taxonomy of each OTU representative sequence was analyzed by RDP Classifier version 2.2 against the maarjam081/AM database using confidence threshold of 70% (*Wang et al., 2007*).

The raw sequence data were deposited at NCBI, in the SRA database with the following accession: PRJNA694003.

## Data analysis

Total soil carbon, nitrogen, and sulphur were determined by an elemental analyser (GC IsolinkFlash 2000; Thermo Scientific, Waltham, MA, USA) analyzer. The concentration of total C, N, and S in soil were 6.96%, 0.55%, and 0.05%, respectively. Meanwhile, the C/N was 12.62. At the same time, we found that the soil temperature was increased 1.4 °C. The dynamic range of the soil temperature was increased from 0.6 °C to 2.4 °C in different elevations. The soil moisture decreased 0.07 $m^3$. The soil moisture increased 0.15 $m^3$ at 4,170 m and decreased 0.11, 0.09, and 0.25 $m^3$ at 3,000, 3,500, and 3,800 m.

The community was expressed by AMF richness, relative abundance and occurrence frequency in different elevations. AMF richness was calculated by the number of OTUs. The relative abundance of AM fungal genus was calculated as the percentage of the sequence number of OTUs in each genus divided by the total sequence number of OTUs in all genera at this altitude. The occurrence frequency of AM fungal genus was defined as the percentage of the number of samples where this genus observed to the number of all samples in this genus. The rate of decrease = (the number of OTUs in CK − the number of OTUs in OTC)/the number of OTUs in CK * 100%. The rate of increase = (the number of OTUs in OTC − the number of OTUs in CK)/the number of OTUs in CK * 100%.

AMF alpha diversity in different elevations were expressed and plotted by the index of Shannon, Ace, and Simpson evenness at the level of OTU by SPASS, Excel and Origin, respectively. The differences of AMF richness and AMF diversity in different elevations were analyzed by two-way ANOVA analysis and Duncan in SPSS 19.0 (*Shi et al., 2019*). We analyze the impact of environmental factors on AMF community after warming through RDA analysis on the platform of the Majorbio Cloud Platform (www.majorbio.com). We standardize the data by flattening according to the minimum number of sample sequences on the platform of the Majorbio Cloud Platform (www.majorbio.com).

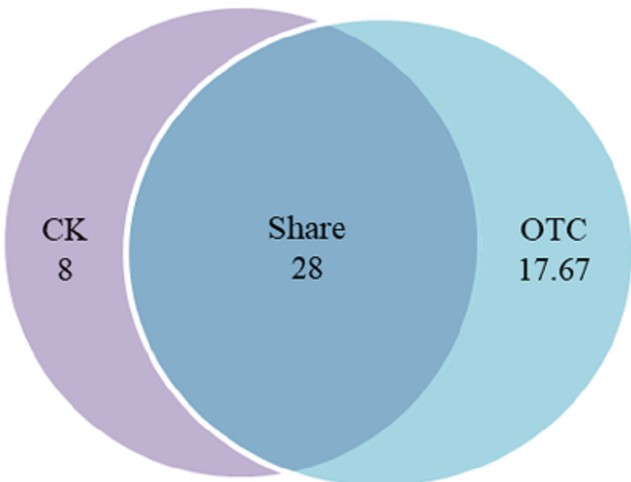

**Figure 1 The influence of warming on AMF richness.** CK means the treatment of control check and OTC means the treatment of warming by open-top chamber. Shared means the treatment of CK and OTC in share. The similarity level was 97%. The data were statistically analyzed by ANOVA (warming: F = 7.509, P = 0.052).

The data of the percentage of relative abundance and occurrence frequency were subjected to square root transformation before analysing and comparing (*Shi et al., 2007*).

## RESULTS

### AMF richness at the level of OTU

Warming increased AMF richness at the level of OTU from 36 to 45.67 with the increase of 26.86%, among them, AMF richness of shared was 28 OTUs, which was 77.78% and 61.31% in CK and OTC, respectively (Fig. 1). In CK, there were 8 unique OTUs, which was 22.22% of the total in CK. AMF richness of shared was 3.5 times to CK solely. In OTC, there were 17.67 unique OTUs, which was 38.69% of the total in OTC. AMF richness of shared was 1.58 times to AMF richness in OTC solely. AMF richness increased but has no significant effects after warming by ANOVA analysis (*P* = 0.052).

### AMF diversity indices at the level of OTU based on different elevations

There were dynamic influences of warming on AMF OTU richness with elevations. OTU richness displayed an upward trend at 3,000, 3,500 and 3,800 m but then decreased at 4,170 m after warming (Fig. 2A). The highest AMF richness occurred at 3,500 m and AMF OTU richness in OTC is greater than that in CK at the elevations of 3,000, 3,500, and 3,800 m, but it was opposite at 4,170 m. That was, AMF richness was lower at the higher altitude after warming. Moreover, elevation had extremely significant effects on AMF richness, which increased significantly at 3,500 m (*P* < 0.001). The interaction of elevations and warming also had a significant effect on AMF richness (*P* = 0.029). The Shannon index has the same tendency to AMF richness (Fig. 2B). At 3,000, 35,00, and 3,800 m, the Shannon index in OTC were higher than that in CK, but showed an

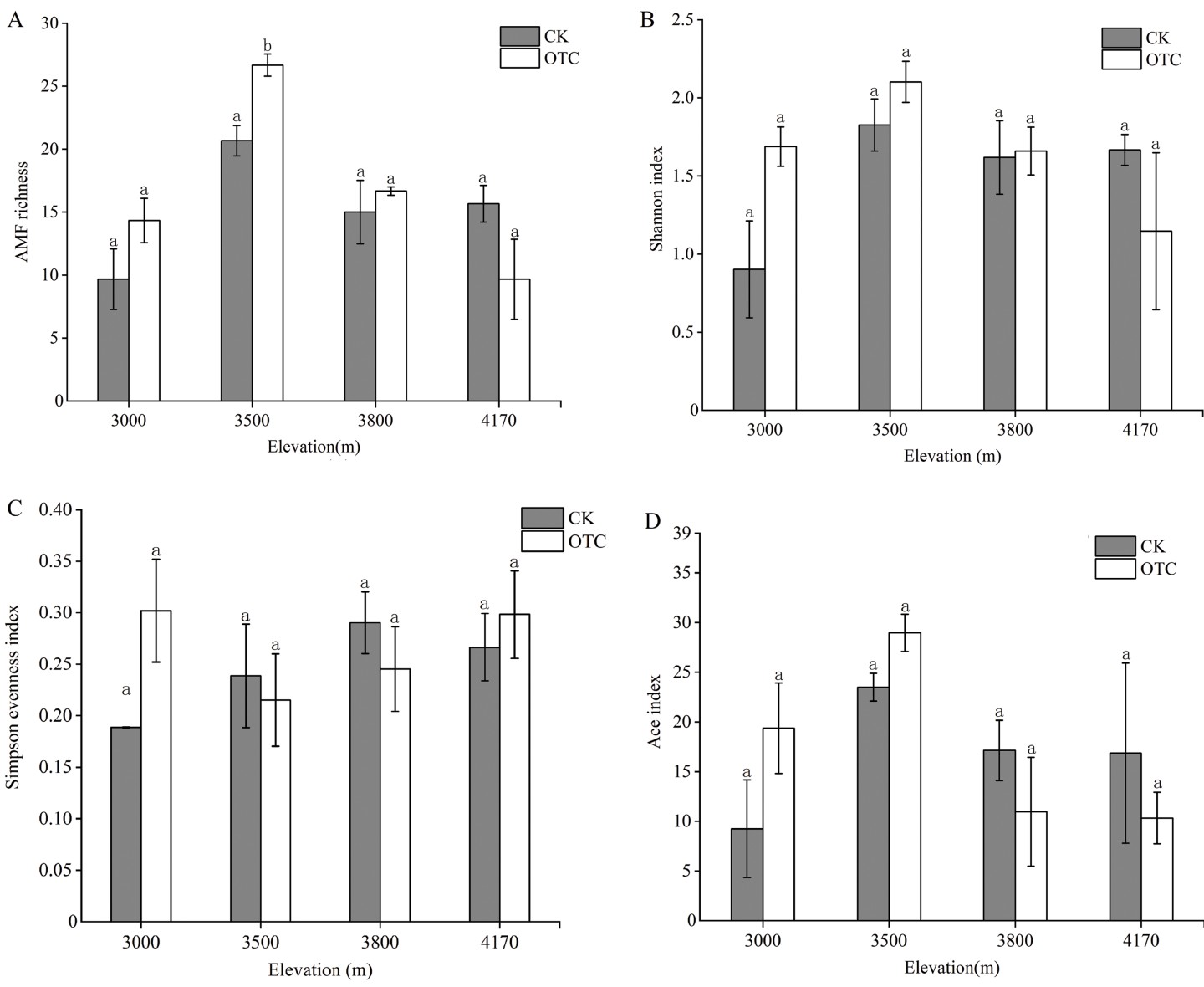

**Figure 2 AMF diversity index at the level of OTU based on different elevations by warming.** Error bars represent the standard error of the mean. Different lowercase letters above each column indicate significant difference, $P < 0.05$. The data were statistically analyzed by two-way ANOVA (elevations: F = 15.387, $P = 0.000$; warming: F = 1.347, $P = 0.263$; elevations × warming: F = 3.874, $P = 0.029$), Shannon index (elevations: F = 2.805, $P = 0.073$; warming: F = 0.682, $P = 0.421$; elevations × warming: F = 2.358, $P = 0.110$), Simpson evenness index (elevations: F = 0.768, $P = 0.529$; warming: F = 0.471, $P = 0.502$; elevations × warming: F = 1.594, $P = 0.230$), Ace index (elevations: F = 3.369, $P = 0.045$; warming: F = 0.047, $P = 0.832$; elevations × warming: F = 1.578, $P = 0.234$).

opposite trend at 4,170 m and none of them are significant. The Simpson evenness index had a similar trend to the Shannon index at 3,000 m (Fig. 2C).

At 3,000 and 3,500 m, the Ace diversity index in OTC were higher than that in CK, but it was opposite at 3,800 and 4,170 m (Fig. 2D). Elevation had significant effects on the Ace index ($P = 0.045$).
Table 2 The influence of warming on AMF richness based on different elevations.

| Order | Family | Genus | Shared | Treatments | | Elevations | | | | | | | |
|---|---|---|---|---|---|---|---|---|---|---|---|---|---|
| | | | | CK | OTC | 3,000 m | | 3,500 m | | 3,800 m | | 4,170 m | |
| | | | | | | CK | OTC | CK | OTC | CK | OTC | CK | OTC |
| Archaeosporales | Ambisporaceae | *Ambispora* | 1.33 | 1.67 | 1.67 | 0 | 0 | 0.33 | 1 | 0.67 | 1 | 1 | 0 |
| | Archaeosporaceae | *Archaeospora* | 3 | 4 | 4.33 | 2.67 | 2.67 | 3 | 1 | 0.67 | 1.33 | 1 | 0.33 |
| | Unclassified | *Unclassified* | 0.33 | 0.33 | 0.33 | 0 | 0 | 0 | 0 | 0.33 | 0.33 | 0 | 0 |
| Diversisporales | Acaulosporaceae | *Acaulospora* | 9 | 10.67 | 12.33 | 4.33 | 4.67 | 6.33 | 8 | 5 | 5 | 7 | 4 |
| Glomerales | Glomeraceae | *Glomus* | 12 | 13.67 | 20.33 | 2 | 4.33 | 9.67 | 15.33 | 6 | 8.33 | 3.67 | 3 |
| Paraglomerales | Paraglomeraceae | *Paraglomus* | 0 | 0.33 | 0.33 | 0 | 0.33 | 0 | 0 | 0 | 0 | 0.33 | 0 |
| Unclassified | Unclassified | *Unclassified* | 1.33 | 5.33 | 6.33 | 0.67 | 2.33 | 1.33 | 1.33 | 2.33 | 0.67 | 2.67 | 2.33 |

Note:
CK means the treatment of control check and OTC means the treatment of warming by open-top chamber. Shared means the treatment of CK and OTC in share.

### The influence of warming on AMF community based on different elevations

Among the genera of *Ambispora*, *Unclassified* (Archaeosporales order), and *Paraglomus*, AMF richness of CK was identical with OTC (Table 2). The largest change in AMF richness was *Glomus*, which increased from 13.67 to 20.33 after warming. For *Acaulospora*, AMF richness was increased from 10.67 OTUs to 12.33 OTUs. The smallest change in AMF richness was *Archaeospora*, which increased from 4 to 4.33.

In addition, there was a downward trend at 4,170 m and the decline rate was 100% in *Ambispora*. However, there was an increasing trend at 3,500 and 3,800 m. For *Archaeospora*, AMF richness was increased at 3,800 m, but decreased at 3,500 and 4,170 m. For *Acaulospora*, AMF richness trended to increase 7.85% and 26.38% at 3,000 and 3,500 m, respectively. As for 4,170 m, it decreased 42.86%. For *Glomus*, AMF richness was increased 116.5%, 58.53%, and 38.83% at 3,000, 3,500, and 3,800 m, respectively, and then decreased at 4,170 m, AMF richness of *Paraglomus* increased at 3,000 m and decreased at 4,170 m. Moreover, the rate of increase on AMF richness at 3,000 m was the same as the rate of decrease at 4,170 m.

The beta-diversity of the AMF community was determined by the Bray–Curtis method (Fig. 3). In the 2-dimensional NMDS plots, soil samples collected from the four different elevations and the two different treatment separated from each other indicating a divergence of the warming treatment. To test the significance, an ANOSIM based on the Bray–Curtis distance showed dissimilarities of the AMF community at the OTU level among the four different elevations and two different treatments ($P = 0.001$).

### The relative abundance and occurrence frequency of AMF

For four different elevations, the relative abundance of *Acaulospora* and *Glomus* were the largest before and after warming (Table 3). The relative abundance of *Acaulospora* and *Glomus* were opposite in four different elevations. At 3,000 m, the relative abundance of other genera showed an increasing trend after warming, except *Glomus* was decreased from 79% to 65%. At 3,500 m, the relative abundance of *Ambispora* and *Glomus* showed an

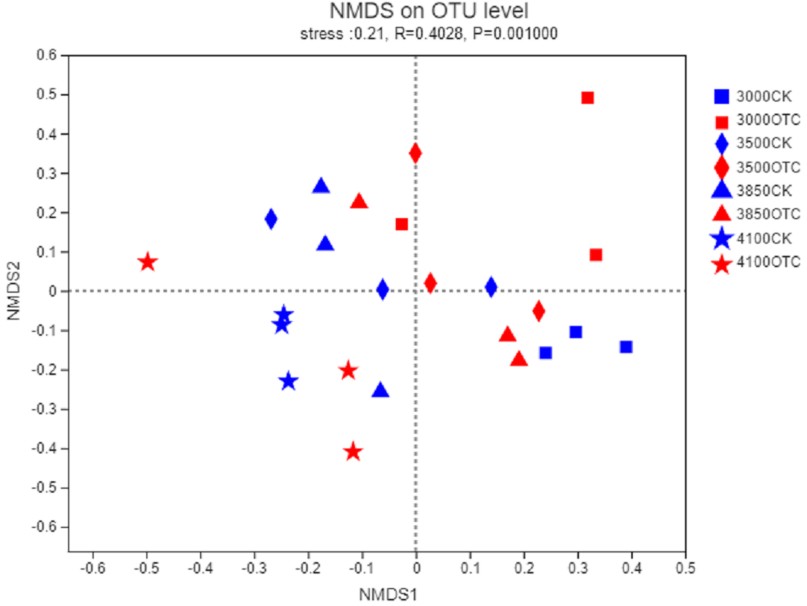

**Figure 3 Nonmetric multidimensional scaling (NMDS) of the influence of warming on AMF community at the level of OTU.** The symbols represent the elevations of 3,000, 3,500, 3,800, and 4,170 m. CK means the treatment of control check and OTC means the treatment of warming with open-top chamber.

**Table 3 The influence of warming on the relative abundance of AMF based on different elevations.**

| Order | Family | Genus | Relative abundance/% | | | | | | | |
|---|---|---|---|---|---|---|---|---|---|---|
| | | | 3,000 m | | 3,500 m | | 3,800 m | | 4,170 m | |
| | | | CK | OTC | CK | OTC | CK | OTC | CK | OTC |
| Archaeosporales | Ambisporaceae | *Ambispora* | 0 | 0 | 0.0490 | 0.0523 | 0.2286 | 0.0294 | 0.3756 | 0 |
| | Archaeosporaceae | *Archaeospora* | 0.8361 | 1.2770 | 0.4115 | 0.05230.1110 | 0.0327 | 0.0555 | 0.5814 | |
| | Unclassified | *Unclassified* | 0 | 0 | 0 | 0 | 0.0621 | 0.0196 | 0 | 0 |
| Diversisporales | Acaulosporaceae | *Acaulospora* | 19.6551 | 32.8304 | 38.9477 | 28.3297 | 37.8633 | 22.0295 | 53.9715 | 55.3139 |
| Glomerales | Glomeraceae | *Glomus* | 79.4761 | 65.1349 | 60.5559 | 71.3894 | 60.8956 | 75.6614 | 44.0003 | 36.4655 |
| Paraglomerales | Paraglomeraceae | *Paraglomus* | 0 | 0.0033 | 0 | 0 | 0 | 0 | 0.0065 | 0 |
| Unclassified | Unclassified | *Unclassified* | 0.0327 | 0.7545 | 0.0359 | 0.1764 | 0.8394 | 2.2274 | 1.5906 | 7.6393 |

**Note:**
CK means the treatment of control check and OTC means the treatment of warming by open-top chamber. Shared means the treatment of CK and OTC in share.

increasing trend after warming, but *Archaeospora* and *Acaulospora* decreased. At 3,800 m, the relative abundance of *Glomus* showed an increasing trend but the relative abundance of *Ambispora*, *Archaeospora*, and *Acaulospora* showed a decreasing trend after warming. As for 4,170 m, the relative abundance of *Archaeospora* and *Acaulospora* increased but *Ambispora*, *Glomus* and *Paraglomus* decreased. The relative abundance of *Glomus* decreased, but *Acaulospora* increased after warming at 3,000 and 4,170 m.

For the four different elevations, *Acaulospora* was always present (Table 4), as was the occurrence of *Glomus*, except at 4,170 m in the OTC treatment. In CK, the occurrence

**Table 4 The influence of warming on the occurrence frequency of AMF based on different elevations.**

| Order | Family | Genus | Occurrence frequency/% | | | | | | | |
|-------|--------|-------|---------|---------|---------|---------|---------|---------|---------|---------|
| | | | 3,000 m | | 3,500 m | | 3,800 m | | 4,170 m | |
| | | | CK | OTC | CK | OTC | CK | OTC | CK | OTC |
| Archaeosporales | Ambisporaceae | *Ambispora* | 0 | 0 | 33.33 | 100 | 33.33 | 66.67 | 100 | 0 |
| | Archaeosporaceae | *Archaeospora* | 100 | 100 | 100 | 66.67 | 33.33 | 66.67 | 100 | 33.33 |
| | Unclassified | *Unclassified* | 0 | 0 | 0 | 0 | 33.33 | 33.33 | 0 | 0 |
| Diversisporales | Acaulosporaceae | *Acaulospora* | 100 | 100 | 100 | 100 | 100 | 100 | 100 | 100 |
| Glomerales | Glomeraceae | *Glomus* | 100 | 100 | 100 | 100 | 100 | 100 | 100 | 66.67 |
| Paraglomerales | Paraglomeraceae | *Paraglomus* | 0 | 33.33 | 0 | 0 | 0 | 0 | 33.33 | 0 |
| Unclassified | Unclassified | *Unclassified* | 66.67 | 100 | 66.67 | 33.33 | 100 | 33.33 | 66.67 | 100 |

Note:
CK means the treatment of control check and OTC means the treatment of warming by open-top chamber. Shared means the treatment of CK and OTC in share.

frequency of *Acaulospora* was the same as that in OTC at different elevations, which seemed that warming had no affect on them. The occurrence frequency of *Ambispora* and *Archaeospora* varied at three elevations, *Paraglomus* varied at two elevations and *Glomus* varied at only one elevations. For different elevations, the occurrence frequency of *Paraglomus* showed a tendency of increasing from 0 to 33.33% at 3,000 m but opposite at 4,170 m. The occurrence frequency of *Ambispora* and *Archaeospora* decreased at 4,170 m. But at 3,500 m, the tendency of *Ambispora* and *Archaeospora* were opposite.

## The influence of soil factors on AMF community by warming

For the four different elevations, RDA1 explained 59.16%, 69.55%, 95.26%, and 95.35% at 3,000, 3,500, 3,800, and 4,170 m of the community structure, respectively. RDA2 explained 0.91%, 2.34%, 1.52%, and 0.56% at 3,000, 3,500, 3,800, and 4,170 m, respectively (Figs. 4A–4D). RDA1 increased from 59.16% to 95.35% with the elevation increased. As the elevations increased, the influence of C, N, S, and C/N were different. C, N, and C/N were positively correlated to RDA1 and RDA2. Sulfur (S) was negatively correlated to RDA1 but positively correlated to RDA2 at 3,000 m (Fig. 4A). At 3,500 m, C, N, and S were all positively correlated to RDA2 but negatively correlated to RDA1 (Fig. 4B). C/N was negatively correlated to RDA2 but positively correlated to RDA1. C and N were negatively correlated to RDA1 and RDA2. S and C/N were negatively correlated to RDA2 but positively correlated to RDA1 at 3,800 m (Fig. 4C). At 4,170 m, C, N, and S were negatively correlated to RDA1 and RDA2. C/N was negatively correlated to RDA2 but positively correlated to RDA1 (Fig. 4D).

## DISCUSSION

The influence of global warming on AMF community structure and the relationship with plant productivity and diversity are important due to the climate change (*Sun et al., 2013*; *Buscher et al., 2012*). Previous studies investigating the influence of warming on AMF usually occurred at a single elevation, which can't accurately reflect the change of AMF community in mountain areas with broad ecosystem topography (*Gai et al., 2012*).

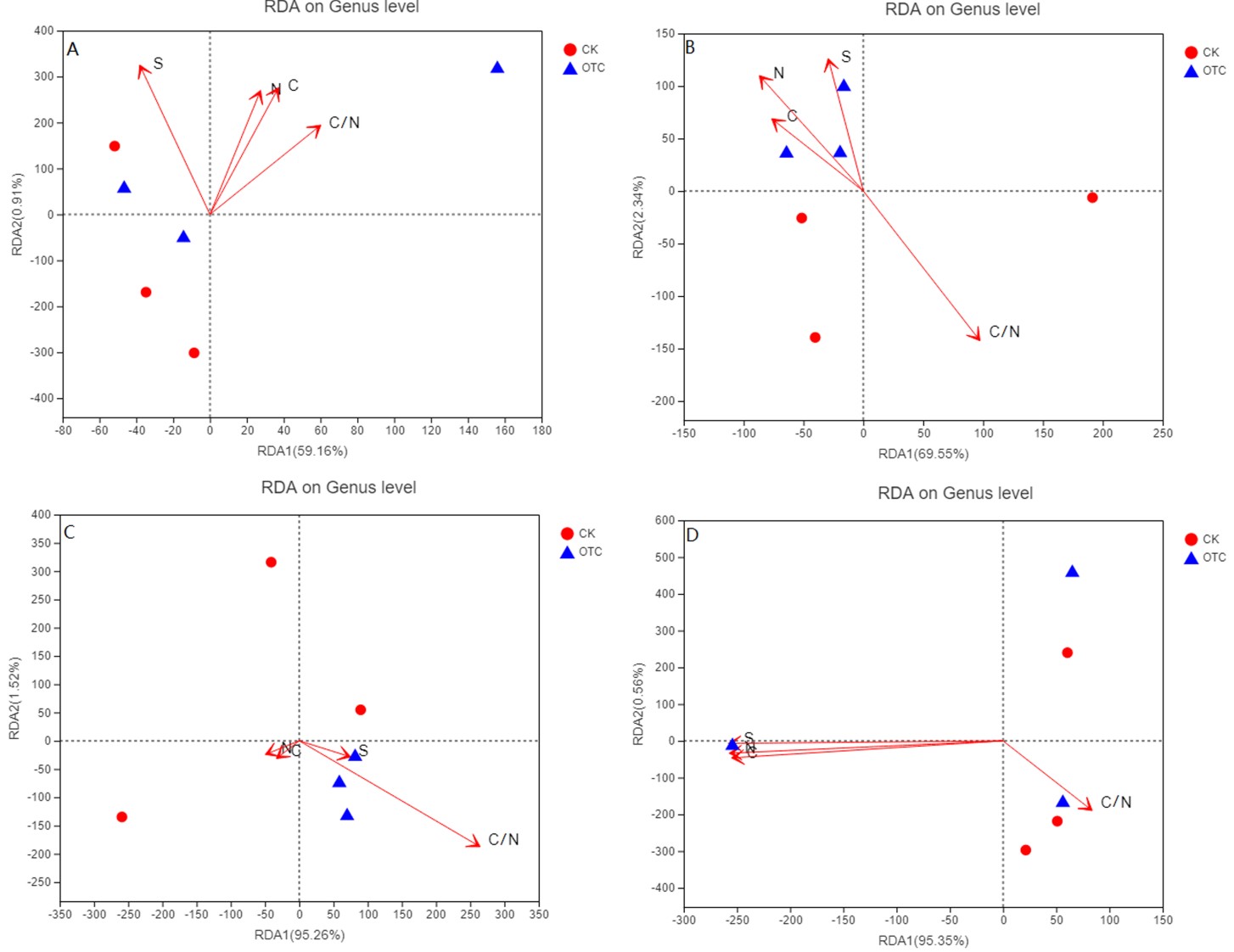

**Figure 4 The influence of warming on RDA analysis at the level of genus based on different elevations.** CK means the treatment of control check and OTC means the treatment of warming by open-top chamber. (A, B, C, and D represent the RDA analysis at 3,000, 3,500, 3,800, and 4,170 m, respectively).

We investigated the influence of warming on AMF community composition at four different elevations by the way of in-situ through open-top chamber on the Qinghai-Tibet Plateau. Our research showed that warming changed the AMF community were dynamic, and varied depending on elevations.

There was also evidence that warming had no effects on AMF community in the semiarid steppe ecosystem (*Gao et al., 2016*). This might be due to AMF communities having little sensitivity to short-term climate change (*Jiang et al., 2018*), or that soil warming had little influence on AMF community were commonly used to seasonal temperature dynamics (*Heinemeyer et al., 2003*). To complicate the matter, AMF

community composition can also be influenced by plant community (*Millar & Bennett, 2016*). In our study, we demonstrated that warming increased AMF richness because there were 17 new OTUs were observed in OTC treatment. Interestingly, as the elevations increased with increasing AMF richness from 3,000 to 3,800 m, but then decreased at 4,170 m. AMF richness was higher at the elevation 3,500 m but not at other elevations. Meanwhile, our results was also similar to *Liu et al. (2016)*, who suggested that warming does not always lead to significant changes in fungal community composition. We reported warming might not influence AMF community compositions at all elevations in this region and had an upper limit then a decline in species richness. AMF community composition response was likely related to the soil moisture and temperature as the soil moisture change was the smallest at 3,500 m whether in CK or in OTC and the soil temperature changed the smallest except 4,170 m (*Gai et al., 2009*). *Sun et al. (2013)* demostrated that soil moisture had an influence on AMF community composition. In addition, there could be an inflection point between 3,800 and 4,170 m, which needs further research.

The analysis of AMF diversity indices including Shannon index, Ace index, and Simpson evenness index in different elevations found that warming had no significant effects on AMF diversity, which had been reported previously (*Gai et al., 2012*; *Yang et al., 2013*). However, the influences of warming on AMF diversity might be varied in different ecosystems (*Kim et al., 2014*). Therefore, we argued that overall (at all elevations) warming had little influence on AMF diversity, which might be related to the ecosystems at different elevations. Indeed, it could be that plant species and elevation exert significant influences on AMF diversity (*Li et al., 2014*), as diversity of the host plants could determine AMF diversity (*Shi et al., 2014*). Therefore, we suggested that the reason of AMF diversity did not increase could be that plant identity had played an overriding role.

Previous studies have shown that there were dominant genera in AMF communities, such as *Glomus* and *Acaulospora* (*Dobo, Asefa & Asfaw, 2016*; *Belay, Vestberg & Assefa, 2013*). Our study found that the relative abundance and occurrence frequency of *Glomus* and *Acaulospora* were higher whether in CK or OTC than other genera except the occurrence frequency of *Glomus* at 4,170 m, which had been reported before (*Sturmer & Siqueira, 2011*; *Coutinho et al., 2015*). *Glomus* had been reported to be dominant in roots according to DNA sequencing, though diversity indices did change between sampling roots, hyphae and soil (*Mickan et al., 2017*). The changes of the relative abundance of *Glomus* and *Acaulospora* were just opposite after warming. The relative abundance of Acaulosporaceae had been shown to increase with elevations increased (*Yang et al., 2016*), which were the same as our study of the relative abundance of *Acaulospora* (Acaulosporaceae Family) at 3,000 and 4,170 m. It appeared that this genus was differed in adaptability at different elevations after warming and there were also differences between different genera. Besides, studies had also reported that *Glomus* and *Acaulospora* were most in different plants, respectively (*Schenck & Kinloch, 1980*; *Blaszkowski, 1989*). Therefore, it indicated that *Glomus* and *Acaulospora* were also highly adaptable to different plants in mountainous areas.

The influence of warming on RDA analysis showed that RDA1 increased with elevation increased. At the same time, C and N were from positively correlated to RDA1 and RDA2 at 3,000 m to be negatively correlated to RDA1 and RDA2 at 3,800 and 4,170 m. C/N had a great effect at 3,000, 3,500, and 3,800 m, but opposite at 4,170 m. The influence of C, N, and S were greater at 4,170 m than that of other three elevations. It indicated that soil factors might change the direction of action on AMF community, though none of them were significant in this experiment.

## CONCLUSIONS

Warming changed the AMF community were dynamic, and these responses varied depending on elevations which consistent with our assumptions that warming significantly changed AMF community structure. Moreover, the occurrences of AMF in different genera also presented the different responses to warming in four elevations. Our results imply that climate change effect of global warming and geographical elevation lead to changes in AMF community, which play an important role in the responses of ecosystem level processes.

### Funding

This work was supported by the National Natural Science Foundation of China (No. 31670499), the Program for Science & Technology Innovation Talents in Universities of Henan Province (18HASTIT013), the Scientific and technological research projects in Henan province (192102110128), the Key Laboratory of Mountain Surface Processes and Ecological Regulation, CAS (20160618), the Training Program for college students (202010464067, 2020337), and the Innovation Team Foundation (2015TTD002) of Henan University of Science & Technology. The funders had no role in study design, data collection and analysis, decision to publish, or preparation of the manuscript.

### Grant Disclosures

The following grant information was disclosed by the authors:
National Natural Science Foundation of China: 31670499.
Universities of Henan Province: 18HASTIT013.
Scientific and Technological Research Projects in Henan Province: 192102110128.
Key Laboratory of Mountain Surface Processes and Ecological Regulation, CAS: 20160618.
Training Program: 202010464067 and 2020337.
Henan University of Science & Technology: 2015TTD002.

### Competing Interests

The authors declare that they have no competing interests. Bede S. Mickan was employed by the Richgro Garden Products at the time the work was conducted.

## Author Contributions

- Mei Yang conceived and designed the experiments, performed the experiments, analyzed the data, prepared figures and/or tables, authored or reviewed drafts of the paper, and approved the final draft.
- Zhaoyong Shi conceived and designed the experiments, performed the experiments, analyzed the data, prepared figures and/or tables, authored or reviewed drafts of the paper, and approved the final draft.
- Bede S. Mickan analyzed the data, prepared figures and/or tables, authored or reviewed drafts of the paper, and approved the final draft.
- Mengge Zhang performed the experiments, prepared figures and/or tables, and approved the final draft.
- Libing Cao performed the experiments, prepared figures and/or tables, and approved the final draft.

## Field Study Permissions

The following information was supplied relating to field study approvals (i.e., approving body and any reference numbers):

Field experiments were approved by the Key Laboratory of Mountain Surface Processes and Ecological Regulation, Chinese Academy of Sciences (20160416).

## DNA Deposition

The following information was supplied regarding the deposition of DNA sequences:

The raw sequence data are available at NCBI SRA: PRJNA694003.

## Data Availability

The raw measurements are available in the Supplemental Files.

## Supplemental Information

Supplemental information for this article can be found online at http://dx.doi.org/10.7717/peerj.11792#supplemental-information.

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
