# Peer review of "Alterations to arbuscular mycorrhizal fungal community composition is driven by warming at specific elevations"

_PeerJ, doi:10.7717/peerj.11792_

## Round 0.1 · original submission · Major Revisions

Reviewers 2 and 3 provide guidance.

Reviewer 1 ·

Basic reporting

Yes

Experimental design

Yes

Validity of the findings

Yes

Additional comments

The manuscript titled “Community structure of arbuscular mycorrhiza fungi are influenced by elevation and the associated temperature changes ” by Yang et al studied the influence of warming and elevation on the community structure of arbuscular mycorrhizal fungi in Qinghai-Tibet Plateau. The results showed that warming increased AMF abundance increased from 57 to 70, but had no impact on AMF abundance. The present results sounds interesting and give the reader some new information on the response of AMF to elevation and global warming. However, the paper is in need of some major improvements before publication.

Firstly, The title should be changed. More description on experiment design should be added, such as, how long the warming time, the measurement of C, N, S, C_N. Warming and elevation were included in this study, the authors should consider the interactive effects, so that two-way ANOVA should be used to analysis the results. Moreover, some sentence in the result section should be moved to methods (eg. Line 243-245).

Secondly, both warming and elevation could affect soil temperature and moisture, which is helpful to understand the changes in AMF community structure, the authors should consider and discuss it. If possible, the author can add this results. Besides that warming and elevation might alter AMF community structure via affecting plant community, how about it in your study?

Thirdly, the author calculated the species composition (Shannon index, Simpson, ACE index, etc), and also analyzed the relative abundance, NDMS might be a good choice to understand changes in the community structure of AMF.

Fourthly, The English language should be improved to ensure that an international audience can clearly understand your text. Such as Lines, 86, 87, 225,

At last, there are some errors in the reference citation, such as, the misuse of author (Line 55-56), some words should be Italic or lower case.

Reviewer 2 ·

Basic reporting

The authors investigated the changes of AM fungal abundance and community structure on the Qinghai-Tibet Plateau (QTP) after a one year warming at 4 elevations (3000 m, 3500 m, 3800 m, and 4170 m). The results were interesting and have great contribution to the understanding of AMF distribution. However, the manuscript was not clearly written and some of it was hard to understand. Several basic information for the study sites were missing, e.g. the mineral nutrient level, the temperature, plant species. PLEASE, revise the manuscript in the introduction and discussion sections, piling up the description of others' researches does not does not validate data and opinion.

Experimental design

The experimental design was clearly stated, but several information were miss.
The measurement of soil proporty, the plant species, the change of soil temperatures was not written in the MM sectio.
Additionally, the repetitions per treatment was 3.

Validity of the findings

no comment.

Additional comments

Line 50-53, 56-58, 370-374, I recommended Keep all references together rather than declaring them separately.
Line 99-100, Please describe soil type and soil properties.
Line 166, Control check - > control check
Line 167, Your means AMF abundance only in CK and OTC, respectively?
Line 205-209, The rate of increase and decrease are confusion. You should write in more detail.
Line 253-254, the relative abundance of Unclassified is zero in CK but not zero in OTC.
Line 304, The description is error, change it.
Table 1-3, Family and genus names should be arranged on one line. I can not find the shared between CK and OTC.
Fig. 2-5, No lowercase letters above each column to indicate significant diffidence, and columns did not start at the origin.

Reviewer 3 ·

Basic reporting

This work investigates modification of AMF communities upon rising temperatures through anin-situ manipulation experiment, using open top chambers, on an altitudinal gradient in the Qinghai-Tibet Plateau.
While the aims of the work and hypothesis made by authors are clear and the background is quickly discussed in the introduction section, results are not clearly presented and discussed. The work seems to suggest that warming can significantly change the AMF community structure, with different variations according to the altitude, it has no effects on AMF abundances.

1. Overall, one of the main issue is the English language which needs a deep revision: in many parts, sentences do not stand (e.g. lines 61-63) and oftentimes are too short and the subject is also omitted (see e.g. line 32).

2.The Introduction needs to include additional and up to date literature on AMF including proper references on their biology and on the induced resistance in plants.

Experimental design

3. I’m really concerned on the data analysis step since data normalization has not been performed, at least as reported in the manuscript. Usually, amplicon sequence abundance data needs to be normalized accounting for sequencing depth. This can be primarily performed by rarefacting the OTU table, or better using rarefaction-free methods such as DESeq2 or other. Please include among methods section few more details on data analysis including downstream steps. As an example I did not understand why in Fig. 1 caption “n=3” is stated. If 3 biological replicates were considered it should be at least n=3 replicates *2 treatments*4 sampling altitudes. Please explain in detail what had been done.

4. I noticed in many parts of the work authors tended to confuse abundance with diversity. As an example, in the first Results paragraph they talk about abundances but shows and discuss the number of detected OTUs, which in microbial ecology are intended as diversity (together with the diversity indexes). I would restructure the Results section accordingly.
Similarly at line 242 the definition of Relative Abundance was completely wrong. I would encourage authors to include an expert microbial ecologist among authors list. Moreover, the effect of warming by OTC on Relative abundances needs to be statistically tested with a proper algorithm.

5. The meaning of “isolation” (from line 241) looks really obscure. Please detail what this parameter is and how it had been calculated since it is not commonly measured in microbial ecology and here is extensively discussed.

6. Overall, methods needs to be integrated in many parts. E.g. it is not clear where samples were harvested and which the features of the sampling site (vegetation cover, soil type, slope and so on..). All these information, including detailed coordinates for each point, should be reported in a supplementary table. Moreover, in many parts, proper references are missing: used methods, including software, primer pairs and databases used needs to be properly referenced. The number of replicates sequenced at each plot (altitude step) and the sampling season should also be clearly mentioned.

Validity of the findings

Unfortunately the statistics is almost fully lacking in a great part of the figures showed. A deep data re-analysis would be required to confirm the trends showed and discussed by authors.

Additional comments

- in the abstract authors stated that “AMF abundance increased from 57 to 70” (line 27). However, few lines below they also stated “But has no significantly effects on AMF abundance” (line 32). Please check.
- line 43: the words “assimilates” is not clear, please give more detail.
- line 57-58: here the citation is already occurring at the beginning of the sentence, please rephrase or add additional evidences rising from both these two papers.
- lines 75-77: this is not fully true, please see these papers: https://doi.org/10.3390/su12145617, https://doi.org/10.1371/journal.pone.0076447.
- lines 111-112: it is not clear here how temperatures were monitored through all the season.
- line 143: if demultiplexing has been performed “in house” the software used needs to be mentioned.
- figure 2 to 5 should be merged into a single one as they show basically the same thing. Moreover, even if in the caption of some of them it is mentioned, statistics is lacking.
- line 196: this sentence do not stand, what is 8? Is this a fold-change? Please give more details and possibily integrate methods section.
- line 226: please check “Maenwhile”

---

## Round 0.2 · Minor Revisions

Please follow the advice of reviewer 3.

Reviewer 1 ·

Basic reporting

The current version improved a lot and addressed all questions in the last version. The revised manuscript can meet the standards of PeerJ.

Experimental design

The experimental design is thoughtful, and the description is sufficient.

Validity of the findings

The results of this manuscript is very interesting, and which could give the reader some new insights on the effect of warming on soil microbes in Alpine grassland ecosystem.

Additional comments

The revised manuscript addressed the questions in the last version, the current version highly improved, which can be accepted for publication.

Reviewer 3 ·

Basic reporting

Since its first version the manuscript has been slightly improved incorporating almost all the comments raised from reviewers. However I still noticed that the language quality is really poor: iI feel like it should be deeply revised by a professional service or a native language speaker.
Moreover, even if many references on AMF were integrated in the introduction section, unfortunately they are mostly outdated. There are a plenty of new references available (see as an example those referenced in this review https://www.nature.com/articles/s41579-020-0402-3#Bib1).
Finally, the references related to primer pairs used to generate amplicon sequencing data are still missing.

Experimental design

Method section has been integrated with many missing information and statistics added. However, the pairwise statistic posthoc test performed after ANOVA should be cited also in Fig.2 caption. Moreover, even if analysis details have been added some further detail is missing. As an example it is not fully clear how reads were processed. Authors mentioned that reads were demultiplexed using bcl2fastq software which actually removes machine libraries barcodes, not sample barcode. However, at line 168, they mentioned that sample barcodes were recognized using FLASH, which however, as far as I known, does not have any demultiplexing options. Finally, the used sample barcodes are not mentioned anywhere.
Again, at lines 198-201, the methods used to perform the analysis mentioned, are not provided. Please integrate all this missing information.

Validity of the findings

no comment

---

## Round 0.3 · accepted · Accept

Thank you for addressing the reviewers' comments.